# Ocean tides trigger ice shelf rift growth and calving

O. J. Marsh [1] ✉, R. J. Arthern[1] & J. De Rydt [2]

Tabular iceberg calving reduces ice-shelf extent, affecting ocean circulation and ice-sheet stability. Here we present detailed observations of a rift on the Brunt Ice Shelf, East Antarctica, from 2017-2023 and its behaviour in the lead up to calving in January 2023. The timing of rift propagation was controlled by the rate of change of ocean tide height, wind speed, and an iceberg collision in August 2021, as well as the long-term ice dynamics. A viscoelastic rheological model is used to estimate the relative magnitude of stresses acting on the rift and to determine a critical threshold for fracture, which was exceeded during a sequence of propagation events in early 2019. The eventual calving on 22nd January 2023 occurred at the peak of a spring tide, supporting the conclusion that tides directly influenced the timing.

Ice shelves are vital for the stability of marine-terminating ice sheets[1]. Sections calve from their seaward margins on multi-decadal time-scales, producing discrete tabular icebergs that impact local ocean circulation and freshwater flux[2,3] and form a key component of the overall mass balance of the Antarctic continent[4]. Irregular or accelerated rates of calving can destabilise ice shelves and reduce their ability to buttress outlet glaciers upstream[5]. Predicting large calving events is important in both ice sheet and coupled earth system models due to the resulting changes in model geometry, ocean properties and melt rate[6,7]. Nevertheless, the processes that drive fracture growth in ice remain poorly observed by field measurements and abstractly represented in models.

On ice shelves, rifts form when there is a change from compressive to extensional stress, usually occurring at the mouth of embayments or on the downstream side of islands or pinning points. Fracture growth is generally considered to be dominated by glaciological stresses[8], but external influences from waves[9,10], tsunamis[11], ocean swell[12], atmospheric extremes[13], sea ice extent[14,15] and iceberg collision[16] have all been shown to cause or provide resistance to calving. In Greenland, tides have been implicated in tidewater glacier calving through vertical growth of crevasses associated with fluctuating water pressure [17,18].

The Brunt Ice Shelf in East Antarctica has seen the growth of several rifts in the last decade and has been studied in detail by the British Antarctic Survey for over 60 years[19]. Local ice dynamics are dominated by the compression of ice into a small bathymetric high point known as the McDonald Ice Rumples (MIR) that is located 60 km downstream of the grounding line (Fig. 1c). The strength of the connection here is fundamental to the velocity of the ice shelf[20,21], with weakening leading to substantial acceleration since calving in early 2023 [22].

In the short term, tides cause semi-diurnal elevation changes of ±2 m and impact ice velocity by up to 100% within one day[23]. Tidally varying flow is observed on ice shelves around Antarctica, particularly strongly in the Weddell Sea[24], and has been explained by the interaction of the ice shelves with grounded regions[25] or varying sea surface slopes[26]. Here, we investigate the effect that tides have on fracture processes by looking at the behaviour of a rift known as Chasm-1. Long-term evolution of Chasm-1 from 2012 to 2019 is attributed to evolving stress near the MIR and spatial variation in ice properties[27–29]. This rift originally formed close to the grounding line in the southern part of the ice shelf and completed its intersection with the coast in January 2023, forming the 1500 km² A-81 iceberg (Fig. 1h).

We monitored the width of Chasm-1 between 2017 and 2023 using four permanent GNSS receivers, which advect with ice flow, and a specially configured ApRES phase-sensitive radar system (Figs. 1, S1). There is no adjustment due to strain rate in intact ice, which is around 0.001 a⁻¹. Within this six-year dataset we focus on behaviour over several shorter periods, including (1) the transition from sustained rift propagation to stagnation between February and September 2019; (2) the response immediately following a collision of iceberg A-74 in August 2021; and (3) the behaviour in the months preceding calving in 2023.

[1]British Antarctic Survey, Cambridge, UK. [2]Northumbria University, Newcastle, UK. ✉e-mail: olrs@bas.ac.uk

## Results

### a – Transition from propagation to stagnation in 2019

Between 25th November 2018 and 20th May 2019, Chasm-1 propagated 4 km in a sequence of discrete events (Fig. 1j). As the rift tip approached the MIR in June 2019, the opening rate decreased sharply, followed by a period of stagnation lasting ~two years (Fig. 1i), although the rift continued to widen. During both propagation and stagnation phases, sub-daily and fortnightly variations in rift opening angle indicate a mixture of recoverable and irrecoverable strain at the crack tip (Fig. 2a, b). We use a 1D viscoelastic model to interpret the observations, where a single time-dependent stress is responsible for both elastic and viscous deformations, which together give the total deformation[30]. In this model, the remote stress field satisfies the shallow shelf approximation for an ice shelf extending uniaxially in the x-direction, perpendicular to the crack[31]. There is no vertical variation in strain rate so observations at the surface are representative of depth-averaged values[32]. We relate deviatoric stress perpendicular to the rift, $\tau_{xx}$ to strain $\varepsilon_{xx}$ using Glen's Flow Law, where A is a viscous rate factor dependent on temperature, and n is the stress exponent[33]. In the viscous part, the stress-strain rate relationship is non-linear. In the elastic part, the stress-strain relationship is linear following Hooke's Law where $E'$ is a material constant and a function of Young's Modulus and Poisson's Ratio:

$$\varepsilon(t) = \frac{1}{E'}\tau_{xx}(t) + A\int_0^t \tau_{xx}{}^n dt \qquad (1)$$

This rheological model is the basis for a relationship between a non-dimensional stress scalar perpendicular to the rift at its tip, $\zeta$, and the local deformation that allows the crack to widen, represented by the rift opening angle, $\psi$ (see Methods):

$$\psi(t) \propto \frac{1}{E'}\zeta(t) + A\int_0^t \zeta^n dt \qquad (2)$$

We determine from our observations of variability in $\psi$ that $\zeta$ can be split into three terms—a constant term $\zeta_g$, a periodic term $\zeta_t$ that is a function of the tidal amplitude, and an irregular term $\zeta_w$ which is a function of the wind speed:

$$\zeta = \zeta_g + \zeta_t + \zeta_w \qquad (3)$$

The opening angle variability is now attributable to combinations of each of the glacial dynamics $\zeta_g$, tides $\zeta_t$ and wind $\zeta_w$, where the tide and wind components are scaled to match independent observations of tides and wind on the ice shelf. In the case of $n = 3$, $\zeta$ splits into three short-term, elastic ($_E$) components and ten long-term, viscous ($_V$) components:

$$\psi = \psi_E + \psi_V$$
$$\psi_E = \psi_{Eg} + \psi_{Et} + \psi_{Ew}$$
$$\psi_V = \psi_{Vg^3} + \psi_{Vt^3} + \psi_{Vw^3} + \psi_{Vg^2t} + \psi_{Vg^2w} + \psi_{Vt^2g} + \psi_{Vt^2w} + \psi_{Vw^2g} + \psi_{Vw^2t} + \psi_{Vgtw}$$
$$(4)$$

Furthermore, because the temporal variability in $\zeta$ drives both the short- and long-term variability in $\psi$, and because there is non-linearity in the long-term behaviour caused by the rate exponent (n) but not in the short-term behaviour due to linear elasticity, the unknown scaling between $\zeta$ and $\tau_{xx}$ can be calculated from the relative magnitudes of the short- and long-term variability in opening angle using published values of A, E, and Poisson's ratio. This allows absolute values of deviatoric stress to be estimated (Fig. 2).

In 2019, the overall opening rate is dominated by a single term— the viscous response due to glacial stress combined with the square of the tidal contributions. The non-linear tidal part creates the strong fortnightly variability (Fig. 2c, d), which is also seen in ice shelf velocity data[24]. The tidal elastic term ($\psi_{Et}$) accounts for most of the sub-daily variability, where narrowing of the rift occurs during falling tides. Four other components in Fig. 2 contribute slightly, while the other seven terms are negligible and not shown. For this rift over this period, the tide-induced component of stress during spring tides exceeds the glacial component due to ice shelf spreading by a factor of more than two (Fig. 2e, f). The wind-induced component is more irregular, although here it exceeds the glacial part when the wind exceeds 20 m.s$^{-1}$.

Five separate rift propagation events of between 50 and 500 m occur in early 2019, observed with TerraSAR-X imagery[34] (Fig. 3). These happen when peaks in wind occur simultaneously with spring tides. The model highlights that the combined tidal and wind effects push the stress above a specific threshold at least once in each of the 5- or 6-day windows when we observe propagation. Our estimate of the deviatoric stress threshold at the time of rift propagation is ~240 kPa, which is within the range of surface tensile stresses of 90–320 kPa found to correlate with the appearance of surface crevasses[35].

After June 2019, the opening rate in both ApRES and GPS datasets decreased by ~40% (Figs.1i, 2b). Our model fit indicates that tidal and wind driven components remain the same while there was a reduction of 25% in the glacial component (Fig. 2f), implying a change in wider ice flow is responsible for rift stagnation. Equivalent ApRES data from 2017 and 2018 shows that when the crack tip was further from the MIR, the maximum deviatoric stresses experienced before failure were still around 240 kPa, although with weaker tidal and wind components and a stronger glacial component (Fig S2). The glacial part is also consistent with previously modelled glacial stresses of up to 140 kPa occurring during earlier propagation events in 2016[29]. Here, the influence of tides increased as the rift extended, becoming dominant around late 2018. Propagation of the rift towards the MIR has the combined effect of reducing the length of the ligament connecting the ice shelf to the proto-iceberg and increasing the moment through which tides and winds drive open the crack (Fig. S4).

### Iceberg collision

On 10th August, an iceberg known as A-74, which had calved from the northern part of the Brunt Ice Shelf in February 2021[13], collided with the remaining ice shelf to the west of Chasm-1. This led to rapid propagation of Chasm-1, growth of a secondary rift from the north (Fig. S5) and additional widening of 7 and 3 m at GNSS baselines TT01-KK00 and NN00-OO000 (Fig. 4g). Evidence of collision is visible in Sentinel-1 satellite imagery (Fig. 4a, b). The energy transferred into the ice shelf was sufficient to create ~7 km of new rift length. This single propagation event was greater than the total rift growth throughout the whole of 2019 due to the interaction of ordinary glacial, tidal and wind stresses, although the collision was still insufficient to cause calving.

### Calving

After the collision and by the end of 2022, the opening rate of Chasm-1 had increased to 1.5 m/day at the NN00-OO000 baseline and 2 m/day at TT01-KK00 (Fig. 1g). Both GNSS baselines continued to show strong spring-neap and daily tidal variability. Calving of A-81 and a smaller unnamed section south of the MIR occurred at around 18:00 UTC on the 22nd January (Fig. 1h). An initial 20 m of movement within an hour confirmed disconnection, which coincided with the fastest rising tide during spring tide (Fig. 5d). Movement away from the ice shelf slowed on the falling tide, resuming during the following rising tide around 07:00 on the 23rd January. The detached iceberg drifted 10 km over the next week, maintaining a strong tidal dependence in drift once free from the ice shelf, with fastest movement during rising tide (Supplementary Movie 1).

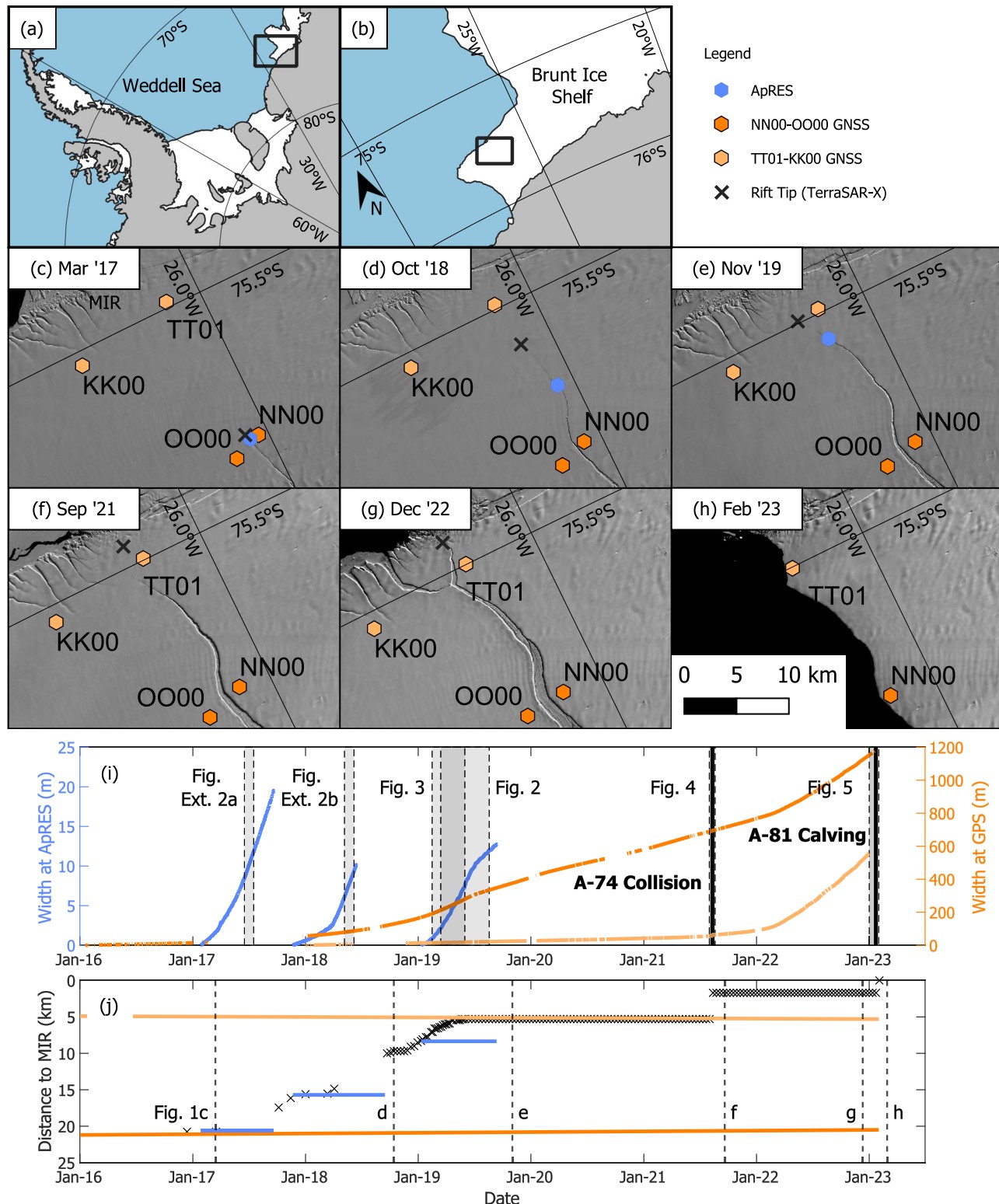

**Fig. 1 | Time series of rift growth on the Brunt Ice Shelf. a** Overview of the Brunt Ice Shelf in the Weddell Sea; **b** the location of panels (**c**–**h**); **c**–**h** Landsat imagery of Chasm-1 from 2017 to 2023, showing the location of instruments and the rift tip; **i** Rift width at fixed GNSS baselines NN00-OO00 & TT01-KK00 (right axis) and an APRES moved annually (left axis), showing the shorter time periods covered by other figures and the timing of A-74 collision and A-81 calving; **j** Satellite-derived positions of the rift tip relative to the McDonald Ice Rumples (MIR), with reference to the timing of panels (**c**–**h**).

## Discussion

The opening rate and timing of propagation of Chasm-1, and the timing of eventual calving of A-81 were strongly influenced by the ocean tide, specifically the rate of change in tide height. The maximum tidal ocean current perpendicular to the rift occurs around 1.5 h after the fastest rate of rising tides in this area[36] synchronous with the observed peaks in rift width (Fig. S6). The correlation in timing implies that tidal variations in ocean drag may be responsible for the widening, rather

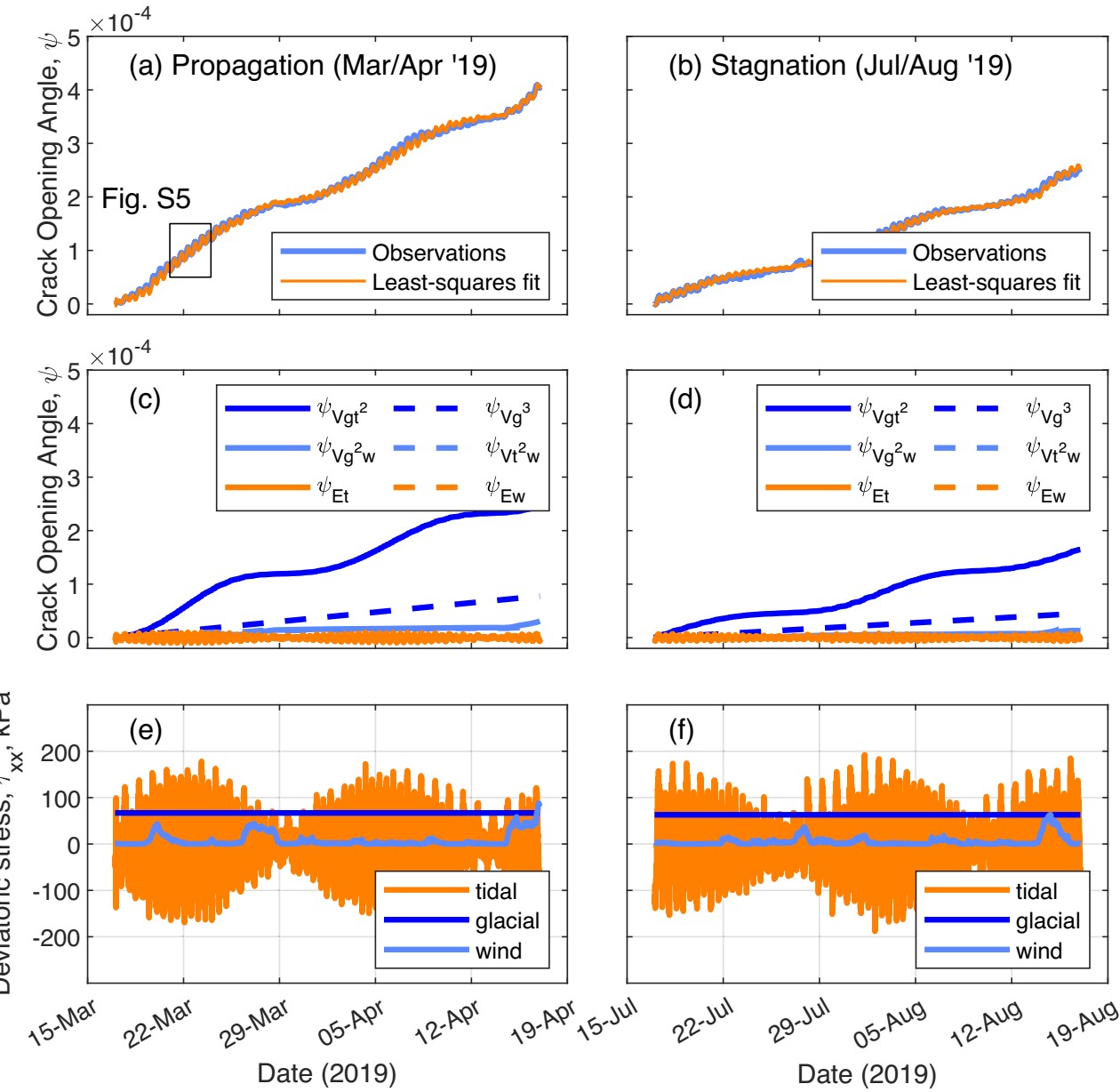

**Fig. 2 | Crack opening angle and estimated stress during 2019. a** during propagation and **b** during stagnation with a least-squares fit to the observations; **c** and **d** the six strongest of the thirteen terms that sum to produce the fitted opening rate ($\psi_{Ew}$ and $\psi_{Vt^2w}$ are small and hidden behind other terms); **e**, **f** a separation of the three forcing terms, converted to stresses at the tip.

than local ungrounding. Amplification of tidal variability as the connection between the iceberg and ice shelf shrinks also supports this mechanism (Fig. S4).

Our simple model excludes wider ice shelf dynamics, including shear around the MIR. We also neglect aspects such as ocean swell, waves and complexities associated with melange forming within the crack, which may help to stabilise crack growth[37]. Nevertheless, a viscoelastic rheology with a constant term, a tidal term and a wind term can be used to explain the short-term variability in crack opening angle, which occurs very clearly above the noise and increases as the crack lengthens. Tides played a dominant role in crack behaviour throughout 2019 and directly influenced the timing of the calving event in 2023. Correlation between high tides, strong winds and five rift propagation events identified from satellite imagery also supports this interpretation. Whilst we have estimated tensile strength from the

maximum deviatoric stress, we have neglected three-dimensional effects and made simplifying assumptions on ice properties and geometry. The tensile strength is also likely to vary spatially and through time under cyclic loading[38], which is not accounted for here.

The sensitivity of rifts on other ice shelves to short-term stresses will depend on their specific geometry, including the relative area of unconstrained ice on which tides and wind can act, and the ligament length attaching nascent icebergs to their ice shelves. There is a large variation in tidal range and rate of change of tide across Antarctica, and we would expect corresponding spatial differences in susceptibility to tidally-triggered fracturing[8]. Tidal currents are large in the southern Weddell Sea and smaller in the diurnal tidal regimes that exist in the Ross Sea, which may explain the pervasiveness of unconfined, seemingly fragile ice tongues in areas of Antarctica such as the Victoria Land Coast [15].

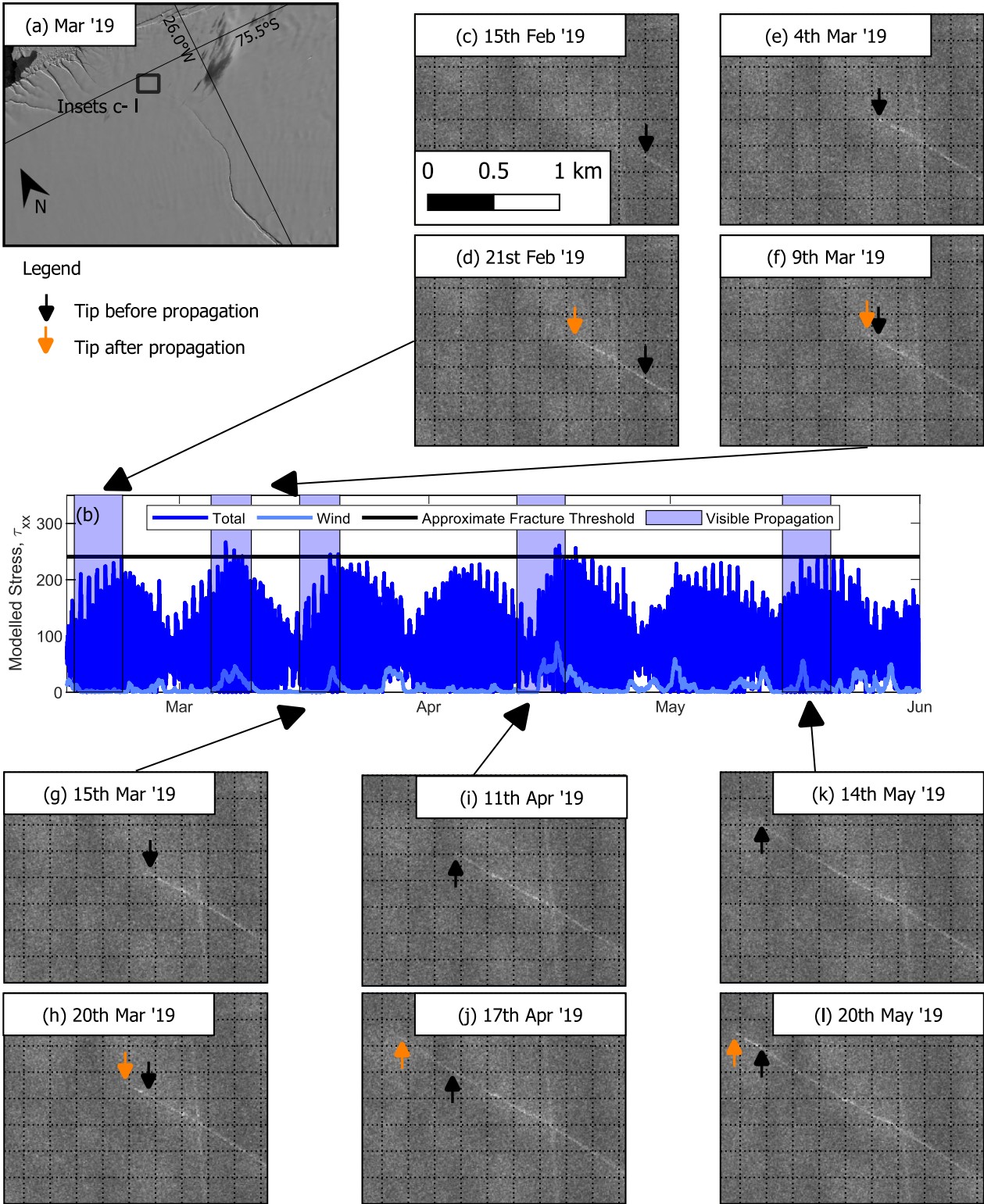

**Fig. 3 | Timing of rift propagation relative to the modelled stress during early 2019. a** Overview of location of the rift tip, the same as in Fig. 1c–h; **b** modelled variability in stress showing that rift propagation coincides with periods when a threshold (here 240 kPa) is exceeded; **c–l** TerraSAR-X backscatter images from before and after each propagation event.

On the Brunt Ice Shelf, we do not observe substantial seasonal variation in rift behaviour that could be associated with sea ice extent, although this cannot be excluded as an additional relevant factor in the final calving, which occurred during the warmest period of the year (late January) and coincided with a sea ice breakout to the south.

The timing of rift propagation is shown to be correlated with rift width, which is itself correlated with easterly tidal currents and easterly winds. Rift propagation occurs repeatedly in discrete events each time a threshold stress is exceeded. For this rift, the fluctuating tidal contribution to the widening rate begins to

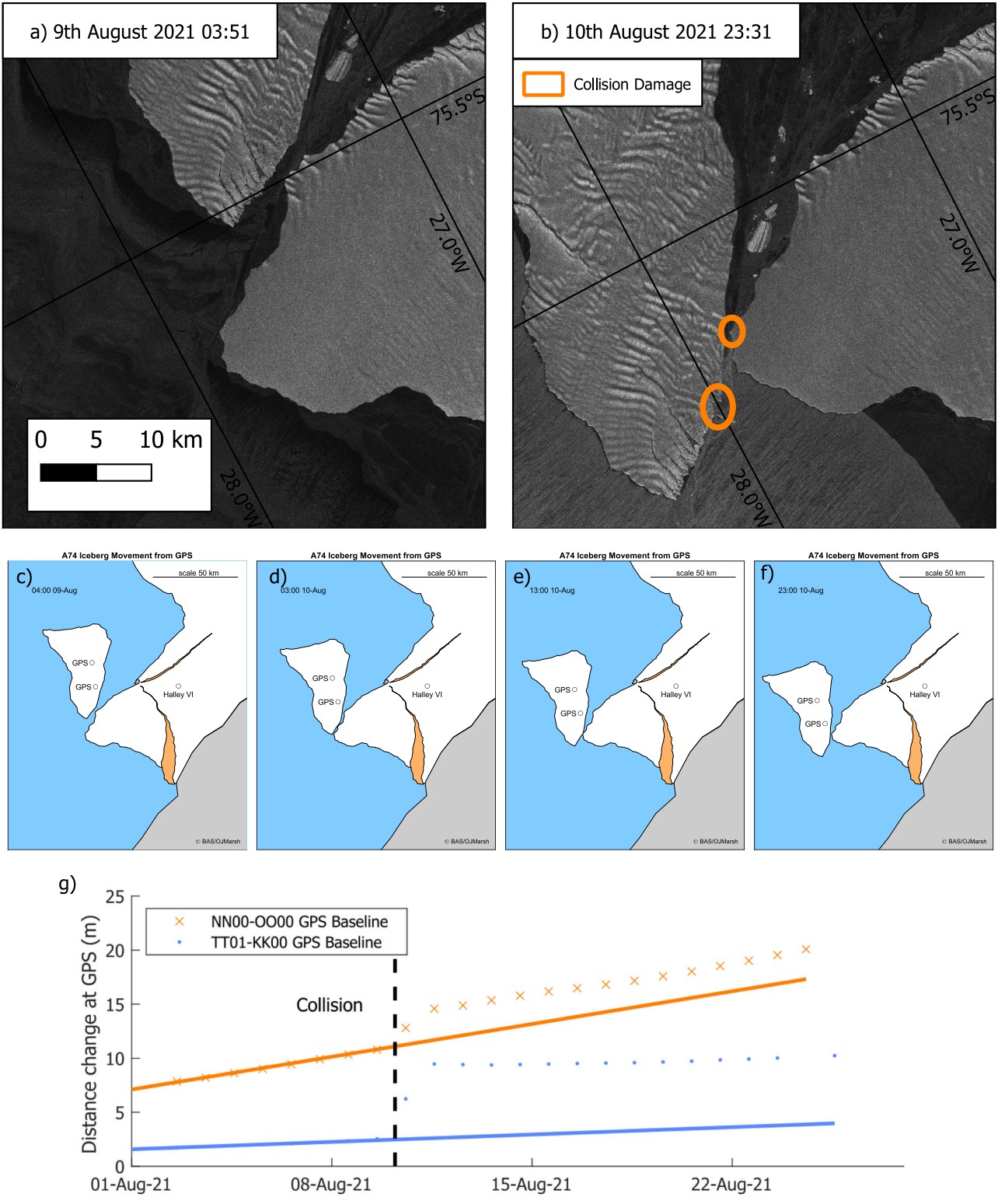

**Fig. 4 | Iceberg collision with the ice shelf in 2021. a**, **b** Sentinel-1 imagery showing collision between A-74 iceberg and the Brunt Ice Shelf; **c**–**f** frames from animation (Supp. 1) showing movement of the iceberg over a period of 2 days; **g** GNSS data showing 7 m of widening of Chasm-1 at the TT01-KK00 baseline on top of a background rate of 0.10 m/day and 3 m of widening at the NN00-OO000 baseline on top of a higher background rate of 0.44 m/day (See Fig. 1g for baseline locations).

exceed the steady glacial creep-induced widening in 2019 and increases further as calving approaches. Strong winds have an additional effect. The tidal widening correlates most strongly with rate of change of the tide and is inferred to be caused by tidal currents creating drag at the ice shelf base. A similar mechanism is proposed for wind drag at the surface.

The unprecedented detail presented here on rift widening and propagation in the lead up to calving is possible due to high-

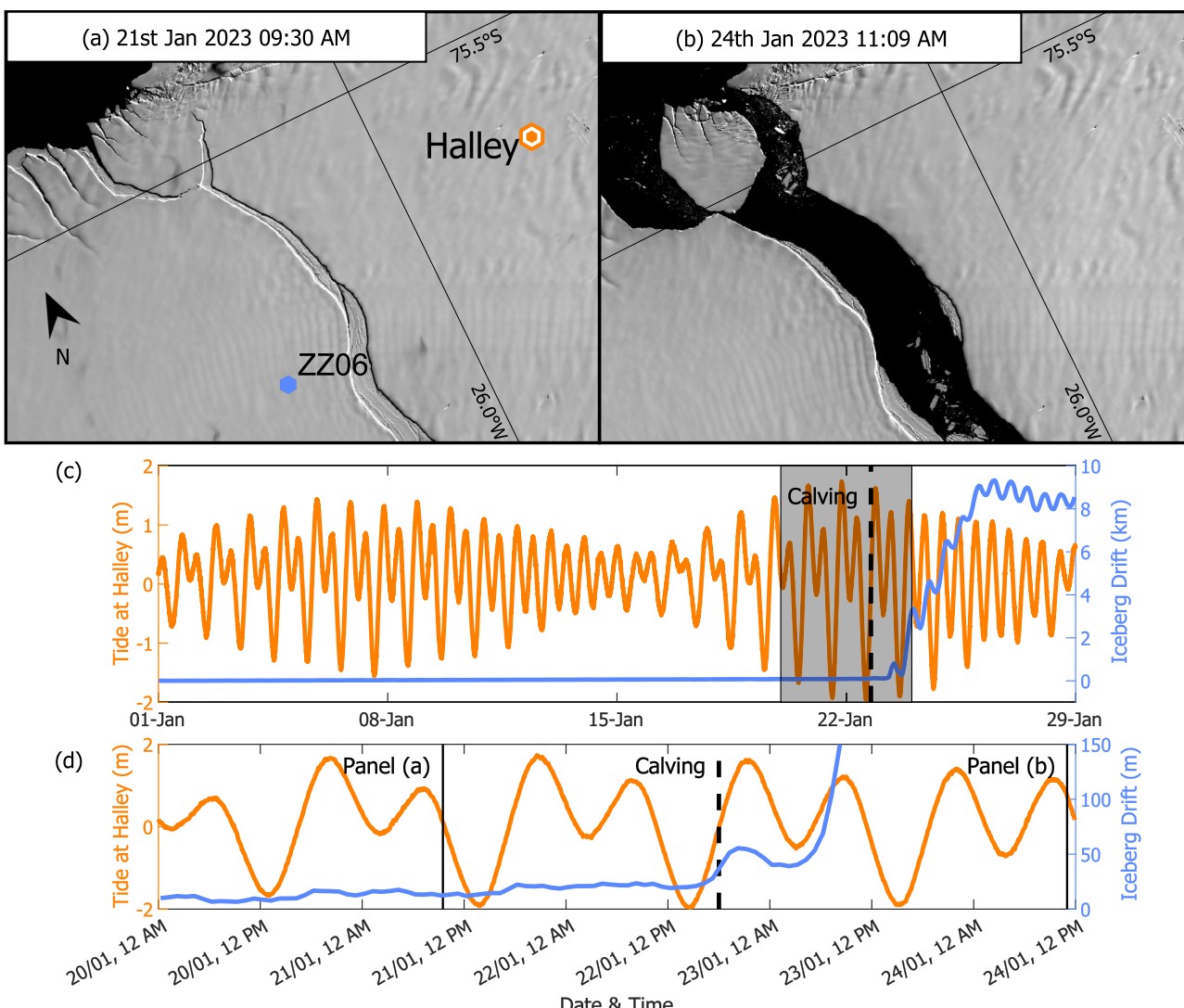

**Fig. 5 | Behaviour in the month leading up to calving. a** the ice shelf before calving from Landsat-9, same region as Fig. 1c-h; **b** the ice shelf after calving from Sentinel-2; **c** the tidal amplitude at Halley (left) and relative displacement of the iceberg at ZZ06 (right) throughout January 2023; **d** the same as (**c**) covering the four days around the calving.

resolution ground-based measurements over multiple years, combined with frequent satellite observations which together allow an understanding of a mechanism not possible through satellite data alone[39]. Tides and wind are key to the timing of small individual rift growth events here, but it is notable that an iceberg collision in 2021 caused more substantial rift growth in a single event than throughout 2020.

Our observations indicate that it should be possible to better predict the timing of tabular calving events from independent environmental data. Identifying the environmental drivers of crack growth is an important step in understanding how ice shelves will respond to changes in the frequency and intensity of extreme atmospheric or ocean conditions in the future.

## Methods
### Field observations
The Chasm-1 rift was monitored by GNSS, satellite and a modified ApRES system[40]. The GNSS receivers at sites named NN00, OO00, TT01, KK00 were Leica GS10s, which recorded their position at 30 s rate for two hours each day. This data was converted into a baseline distance using RTKLIB software[41]. The rift tip location from September

2018 was determined from TerraSAR-X imagery where a strong backscatter contrast between the rift (high backscatter) and undamaged ice (low backscatter) indicates the tip position even when the rift is bridged by drifting snow[34]. Landsat-8 imagery constrains the tip location during austral summer before 2018.

The ApRES was configured with the transmitter and receiver on the eastern side of the rift, pointing horizontally across the rift, and a large metal reflector (2 × 4 m aluminium mesh) on the western side, initially separated by around 30 m (Fig. S1). This produced a strong reflection, which is converted into range assuming a velocity in air of 0.3 m/ns. The system recorded data every 10 min. The uncertainty on the ApRES data (-±2 mm) is substantially lower than the daily tidal changes in width (±50–100 mm). We use a Savitsky-Golay filter to remove noise. The width is converted into opening angle using rift tip positions from the TerraSAR-X backscatter images, labelled manually.

### Rheological model
The model is based on a Maxwell rheology where elastic strain, $\varepsilon^E$, and viscous strain, $\varepsilon^V$, sum to give total strain, and elastic stress $\sigma^E$ is equal to viscous stress $\sigma^V$ Elastic stress is related to elastic strain through

Hooke's Law:

$$\boldsymbol{\sigma}^E = \lambda \, \mathrm{tr}\left[\boldsymbol{\varepsilon}^E\right] I + 2\mu \boldsymbol{\varepsilon}^E \tag{5}$$

Also written as:

$$\boldsymbol{\varepsilon}^E = \frac{1}{2\mu}\left(\boldsymbol{\sigma} - \frac{\lambda}{3\lambda + 2\mu}\mathrm{tr}[\boldsymbol{\sigma}]I\right) \tag{6}$$

Where $\lambda = \frac{E\nu}{(1+\nu)(1-2\nu)}$ and $\mu = \frac{E}{2(1+\nu)}$ are the Lamé parameters, and E and ν are Young's Modulus and Poisson's ratio. We separate the depth-dependent component of $\boldsymbol{\sigma}$ as follows [32]:

$$\boldsymbol{\sigma} = \mathbf{R} - \rho g(s - z)I \tag{7}$$

Here $\mathbf{R}$ is a resistive stress, which can be defined from principal strain rates in x and y directions, $\dot{\varepsilon}_{xx}$ and $\dot{\varepsilon}_{yy}$, and viscosity, η using the shallow-shelf approximation [42]:

$$\mathbf{R} = \begin{bmatrix} 4\eta\dot{\varepsilon}_{xx} + 2\eta\dot{\varepsilon}_{yy} & 0 & 0 \\ 0 & 4\eta\dot{\varepsilon}_{yy} + 2\eta\dot{\varepsilon}_{xx} & 0 \\ 0 & 0 & 0 \end{bmatrix} \tag{8}$$

$\mathbf{R}$, like $\boldsymbol{\sigma}$, can also be split into deviatoric and volumetric parts:

$$\boldsymbol{\tau} = \boldsymbol{\sigma} - \frac{1}{3}tr[\boldsymbol{\sigma}]I = \mathbf{R} - \frac{1}{3}\mathrm{tr}[\mathbf{R}]I \tag{9}$$

So that the deviatoric stress tensor, $\boldsymbol{\tau}$ is:

$$\boldsymbol{\tau} = 2\eta \begin{bmatrix} \dot{\varepsilon}_{xx} & 0 & 0 \\ 0 & \dot{\varepsilon}_{yy} & 0 \\ 0 & 0 & -\left(\dot{\varepsilon}_{xx} + \dot{\varepsilon}_{yy}\right) \end{bmatrix} \tag{10}$$

We then simplify by restricting the ice to uniaxial strain perpendicular to the crack so that strain in the crack-parallel direction (y) is zero. This allows us to write the elastic part of the strain in terms of deviatoric stress as:

$$\varepsilon_{xx}^E = \left(\frac{1}{2\mu} + \frac{1}{3\lambda + 2\mu}\right)\tau_{xx} \tag{11}$$

The constant $\left(\frac{1}{2\mu} + \frac{1}{3\lambda + 2\mu}\right)$ is referred to as $1/E'$. For clarity, $E' = 1.7 \times E$ when $\nu = 0.3$.

For the viscous part, uniaxial strain rate means viscosity, η is controlled only by $\tau_{xx}$ and the rate factor A:

$$\eta = \frac{1}{2}\left(A\tau_{xx}^{n-1}\right)^{-1} \tag{12}$$

So that for the viscous part, the uniaxial strain rate is:

$$\dot{\varepsilon}_{xx}^V = A\tau_{xx}^{n} \tag{13}$$

We now have a single stress component that influences both the elastic and viscous behaviour:

$$\varepsilon_{xx}(t) = \varepsilon_{xx}^E + \varepsilon_{xx}^V = \frac{1}{E'}\tau_{xx}(t) + A\int_0^t \tau_{xx}^n dt \tag{14}$$

As we do not directly measure a strain field ahead of the rift, or an applied stress, we must now introduce two scaling constants. This requires two further assumptions—firstly, the width of the rift is linearly proportional to strain at the crack tip. This is only reasonable when

the tip is stationary and so to account for propagation events, we convert width, w, taken from ApRES measurements into the opening angle, $\psi$, using distance between the ApRES and the current crack tip, d, (Fig. S4):

$$\psi = \frac{w}{d} \tag{15}$$

$$\varepsilon = C_1\psi \tag{16}$$

This minimises any direct feedbacks between propagation and rift width. Secondly we assume that the remote stress on the ice shelf is linearly proportional to the stress that controls the fracture events:

$$\tau = C_2\zeta \tag{17}$$

We split this remote stress into three parts. The glacial term $\zeta_g$ represents the combined effect of imbalance of hydrostatic forces within the shelf, which cause horizontal spreading and tensile stresses induced by flow of the ice shelf. In the model, this is held constant over short time periods of up to a month while we focus on the short-term behaviour.

The tidal term $\zeta_t$ represents the effect of periodic ocean tides on the ice shelf. Tidal amplitude measured with GNSS at the nearby Halley Station 10–15 km is used from 2019 onwards[22], while a tide model is used for 2017 and 2018 data[36]. The GNSS data accounts for the inverse barometer effect on sea level that is not captured in the tide model. The mechanism through which the tides influence the crack width is not directly modelled here, but there is a strong correlation between opening angle and rate of tidal amplitude change (Fig. S6), with rising tides leading to stronger widening. We find a time-lag of around 1.5 h between the tidal signal and the opening angle and use the rate of change of amplitude offset by 1.5 h as a proxy for tidally induced variability in stress. The relative contributions of the tide, wind and glacial components are unknown a prior, and so two further scaling constants are introduced $C_3$ and $C_4$:

$$\zeta_g = 1 \tag{18}$$

$$\zeta_t = C_3\frac{dz(t - \omega)}{dt} \tag{19}$$

$$\zeta_w = C_4 u_w^2 \tag{20}$$

Where z is the detrended height above sea level of the ice shelf surface at Halley (a proxy for tidal height), $u_w$ is the wind speed and $\omega$ indicates the time lag between the tide and the response at the rift.

The wind term $\zeta_w$ represents the strength of easterly winds caused by low-pressure systems to the north. These are common on the Brunt Ice Shelf with wind stresses increasing proportionally to the square of the wind speed[43]. Wind data are taken from a 2 m weather station at Halley Station.

Assuming $n = 3$, this model can be expanded to give a relationship between opening angle and remote longitudinal stress, with thirteen forcing terms that sum to produce the opening angle:

$$C_1\psi = \frac{C_2}{E'}\left(\zeta_g + \zeta_t + \zeta_w\right) + AC_2^3\int_0^t \left(\zeta_g^3 + \zeta_t^3 + \zeta_w^3 + 3\zeta_g^2\zeta_t \right.$$
$$\left. + 3\zeta_g^2\zeta_w + 3\zeta_t^2\zeta_g + 3\zeta_t^2\zeta_w + 3\zeta_w^2\zeta_g + 3\zeta_w^2\zeta_t + 6\zeta_g\zeta_t\zeta_w\right)dt \tag{21}$$

Each of the parameters $C_{1-4}$ controls an aspect of the final shape of the opening angle curve. We vary all parameters together in a four-

dimensional space, then calculate the least-squares fit of each combination relative to the observations, such that the overall minimum indicates the optimal parameter values.

$C_3$ controls the relative magnitudes of the tidal and glacial parts and can be constrained from the viscous behaviour (the relationship between linear rate of opening and spring-neap variation). $C_4$ determines the relative magnitudes of the wind and tidal parts and can be constrained from the elastic behaviour during strong winds, relative to the elastic tidal behaviour. $C_1$ and $C_2$ depend on the relative amplitudes of the viscous and elastic behaviour (Fig S3) and are highly sensitive to the choice of $E$ and $A$ (Fig. S7). The range of values stated in the literature for $E$ and $A$ is large and leads to large variations in the apparent fracture threshold. Here, we use $E = 9$ GPa[44], $\nu = 0.3$ and $A = 5.3 \times 10^{-25}$ s$^{-1}$ Pa$^{-3}$ [45], which are reasonable estimates for ice at $-{-}10$ °C on the Brunt Ice Shelf[19]. The derived value of $C_2$ then allows us to approximate the depth-averaged deviatoric stress threshold for fracture.

## Data availability

ApRES and GPS data collect as part of this research are available via the NERC Polar Data Centre https://doi.org/10.5285/3d6fd5ae-e94d-4d0b-a5e9-5779ca84855e. GPS data from Halley Research Station are available at https://doi.org/10.5285/76dec018-3ea8-4ecc-9b53-5dd915daf214 Wind data from the Brunt Ice Shelf are available at basmet.nerc-bas.ac.uk/sos. TerraSAR-X satellite data are available on request from the ESA archive https://earth.esa.int/eogateway/catalogue/terrasar-x-esa-archive.

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

## Acknowledgements

We would like to thank David Vaughan, Jakub Stocek and others for valuable initial discussions on dynamics of rifts on the Brunt Ice Shelf. We would also like to thank staff of Halley Research Station over several years for help in maintaining instruments on the Brunt Ice Shelf, and for retrieval of the ApRES in 2020-21. TerraSAR-X data in Fig. 3 and S5 were provided by the German Aerospace Centre (DLR) through project HYD_2997. Landsat-8 and -9 images in Figs. 1, 3 and 5 are courtesy of the U.S. Geological Survey. Sentinel-1 imagery in Fig. 4 and Sentinel-2 imagery in Fig. 5 were provided by Copernicus under license CC BY-SA 3.0 IGO. O.J.M and R.J.A were supported by the Natural Environment Research Council through grants NE/X014991/1 and NE/X011372/1.

## Author contributions

O.J.M. and J.D.R. conducted field measurements. O.J.M. processed and analysed data and wrote the manuscript. R.J.A. and O.J.M. derived model equations. O.J.M., R.J.A. and J.D.R. revised the manuscript.

## Competing interests

The authors declare no competing interests.
