## [Transparent Peer Review file · Nature Communications]

Ocean tides trigger ice shelf rift growth and calving

Corresponding Author: Dr Oliver Marsh

Version 0:

Reviewer comments:

Reviewer #1

(Remarks to the Author)

This manuscript describes results from a field campaign on the Brunt Ice Shelf showing that rift propagation appears to be, at least partially, triggered by ocean tides and winds. This study falls into a larger cannon of studies that seek short-term environmental triggers of rifting propagation and iceberg calving. This study is the first that I am aware of that demonstrates a link between rift processes and tidal forcing. One of the challenges, however, of this type of study is that it is difficult to generalize: we do not know if the correlation with tides holds more generally for rifts in the Brunt ice shelf, much less other rifts on other ice shelves. This issue is an issue that has been difficult to overcome in many studies, but doesn't detract from the study presented here.

Overall, I think the study is well written, adds to the literature and will be interesting to a wider audience. However, the methodology and assumptions were difficult for me to fully follow: I struggled to understand and convince myself that some of the assumptions described in the methods are appropriate and link the empirical models to physical models. I recognize that there is a balance that needs to be met with not being overly technical so that the manuscript is accessible to a wide audience and adding more specialized jargon. But I think that some more precision in terminology and clarity in the methods might be useful for readers. This is especially true when asking readers to make the link between analogies and derivations which seem based a little bit more on insight and intuition than formal manipulations. I outline some of these issues below, recognizing that it is possible that some of these are just wording issues while others might point to issues with the methodology that need to be addressed.

1. Cauchy stress, deviatoric stress and all that. To start, the authors use the term "stress" multiple times throughout the study (e.g., line 69, line 74, but also many other places). Stress is, of course, a tensor, but it is treated in the text as if it is a scalar. This is fine so long as the authors are clear that they are referring to a particular component of the stress tensor (or better yet, the traction). I also assume that the "stress" referred to is depth averaged in some sense, but this is also not stated. The more confusing problem is that the authors do not distinguish between Cauchy stress, deviatoric stress and effective stress. In fracture mechanics, the stress driving crack propagation is the Cauchy stress, but flow of ice is controlled by the deviatoric stress. This is an important distinction because the Cauchy stress can be compressive because of the hydrostatic pressure whilst the deviatoric stress remains positive. More specifically, the effective viscosity in the creep flow-law is controlled by the effective deviatoric stress, which is independent of pressure and depends on the sum of the squares of the components of the stress. This is an important distinction because the effective stress cannot be negative and depends on more than a single component of the deviatoric stress tensor. I would typically write the Equation that appears in lines 69 and 74 in terms of the effective stress (which is always positive). All of this is very confusing to readers, especially since the tidal stress shown in Figure 2 has negative values. One has to be careful (and clear) about absolute values and what is positive and negative! Here the authors would better serve readers by clearly stating what they mean by "stress" in a more precise way (it is a component of the full stress tensor? is it depth-averaged? is it deviatoric stress or a component of the Cauchy stress).

2. Local fracture stress, remote stress and rift opening angle. Putting aside the terminology issues, the Equation in Line 69 is valid for a 1-dimensional visco-elastic material. I don't follow the subsequent assumptions and approximations made to then transform the rheological equation in Line 69 to rift opening angle in Line 78. This seems like an analogy then a derivation, which is fine but should be stated. If we consider uni-axial loading transverse to the rift and ignore all other components of

the stress tensor, then it seems like we should be able to write an equation in the form of the Equation on Line 69 with $\epsilon(t)$ denoting the component of strain transverse to the rift. The integral of the strain would be the rift opening/displacement. The rift opening angle used here appears to be defined as the ratio of the width to the length of the rift and this is, at least dimensionally, equivalent to a shear strain. This adds confusion about which component of the stress is considered. It would be helpful to clarify this, but either way, I don't see why the factors of A and E in the Equation in 74 has been dropped. Maybe the ratio is used to scale the rift opening to rift length, but rift width in an elastic material should scale with rift length. Moreover, the rheology is a statement about material properties and has nothing to do with the various stress balances in the far-field or fracture regions. The magnitude of stresses might vary spatially and in the near vs far field, but the rheology is not a statement about the stress balance. Hence, it is not clear to me why we need both equations or why they are written in different forms.

3. Fracture mechanics and other stuff. The methodology sketched out in Lines 218-226 argues for a relationship between the local stress near the crack tip and the far-field stress that is based on fracture mechanics. As I understand it, I think the authors are trying to infer the stress near the crack tip based on the far field stress, which is perfectly reasonable for an elastic material. I can follow the scaling presented, but given the relationships in Equations near lines 218-223 and the fact that the rift length is known, why not use the rift length in the analysis? The proportionality is given by a factor that is related to the rift length and another factor this typically of order unity (often π or some such). At least for an order of magnitude estimate, we could directly estimate the stress associated with different forcing without introducing arbitrary constants. In fact, an alternative interpretation of the analysis is that if we divide both sides of the Equation in line 218 by the square root of the rift half-length, then we find that the remote stress required to drive rift propagation is approximately zero given the critical stress intensity factor from lab measurements. Hence, whenever the Cauchy stress (not deviatoric stress!) is positive, the rift will propagate. This relationship only holds for linear elastic materials, which gets me into my next point.

4. Viscoelasticity and Hoff's analogy. As I understand it, the analogy pointed out by the authors holds for linear viscosity and linear elasticity (or power-law elasticity and power-law creep), but not between linear elasticity and power-law creep. In fact the stress singularity associated with power-law creep has been deduced and it depends on the flow-law exponent. (The singularity is exactly the same as for an elastic material if the flow law exponent is unity.) As a consequence, I don't think that the scaling argument that the authors apply follows. Or, at least, more clarification is needed to explain why it works. The way that the authors present the methods make it seem like this is a critical flaw in the methodology or something that needs additional justification. (I apologize if I misunderstood!) However, reading through a few more times I'm not quite sure that the authors analysis really depends on this scaling argument and I wonder if the authors can simply use the rheology presented in lines 69 with some appropriate measure of transverse strain and simplify much of this exposition.

5. My penultimate concern about the methodology is, perhaps, more of a philosophical issue. The main conclusion is that tides and, to a lesser extent winds, drive rift propagation. The concerns that I raised are all based on the physical model presented, but the authors conclusions are fundamentally based on an empirical correlation with 4 tuned coefficients (5 if the phase of the tide is included) that are fit to the data. However, we can, at the very least, provide ballpark estimates of all of the stress terms based on the data collected. The drag associated with tides and winds depends quadratically on the current/wind speed, density of water/air and a drag coefficient. The wind speed is measured. The current speed can be extracted from the same tidal model used to estimate ocean height. The only unknown is the drag coefficient, which can vary over several orders of magnitude. Nonetheless, the authors can ballpark estimate the stress associated with different forcing using a range of published drag coefficients to determine if the relative contributions inferred empirically are broadly consistent with what we expect from a more physical model. If the forcing is broadly consistent, then it is a strong check on the more empirical methodology and might negate my confusion about the methods. Moreover this might even allow us to estimate the role that winds and tides have on other rifts. This would be a significant advance in the field. If the relative magnitudes are not consistent then it implies drag coefficients need to be much larger/smaller than typically inferred or something else is going on that needs additional study. Perhaps the Brunt is unusual or there is missing physics we still need to understand. This is also important to know.

6. My final point is that it seems as though the authors are using rift widening as a proxy for rift opening. Rift widening in an elastic material is either due to a change in the stress opening the rift and/or rift propagation. In a viscoelastic material the situation is more complicated because the rift will widen due to ductile flow even with no rift propagation. In an elastic material, rift widening results in stored elastic energy that can be released by rift propagation. In a disco-elastic material, rift widening may dissipate stored elastic energy or respond passively to rift propagation. This confusion may be addressed in the manuscript, but it seems possible that the rapid rift widening stages observed are caused by rift propagation, which stand in contrast with rapid rift widening causing stress accumulation that then results in rift propagation. In other words, the direction of causality isn't clear to me. Can the possibility that rapid rift widening is simply a response to the fact that the rift is or has recently propagated?

Minor comments:

A key finding is that the square of the tidal forcing results in fortnightly forcing. Recent studies, however, have increasingly

shown that on ice shelves the flow-law exponent is probably closer to 4. Would this significantly affect the results? Would the fortnightly tidal affect still appear? Does a flow law exponent of 4 help with the tidal phase offset needed in the fit? If the fortnightly forcing is robust, then this seems like a useful prediction of the model based on fundamental principles that is worth further emphasizing.

Is it possible that the connection with wind speed is related to small movements of the stations of the ice associated with torque from the wind?

Line 84: This is the results of expanding the cubic, right? Why are the factors of 3 and 6 in the terms dropped? Line 265: This equation includes all of the factors of 3 and 6s that was missing in the main text equation. It might be easier for readers if these equations are presented in a consistent form.

Line 100: kts—use metric units.

Why do tides result in a phase shifted stress? This isn't intuitive to me. Is it somehow related to the phase difference between tidal currents that drive propagation and tidal heights?

Reviewer #2

(Remarks to the Author)

This is an excellent, interesting and important paper that reports highly detailed observations of rift propagation and iceberg release from Brunt Ice Shelf in Antarctica, and an insightful analysis of the key forces at work. The methods are rigorous and clearly explained and the conclusions justified. Illustrations are excellent.

I see no need for any revision of the text, except for one minor correction. On line 41, there is mention of "velocity fluctuations of plus/minus 2 metres", which should of course be elevation fluctuations, which correspond with velocity fluctuations. It might also be useful to add the units for 'rift opening angle' which I presume are radians (line 72 and Fig 2).

The main change that I think might be useful is to the title of the paper. The title only mentions tidal forcing as the trigger for both rift propagation and calving, but it is clear from the text that this is not the whole story. While rising tidal stresses clearly coincided with individual rift propagation events, iceberg collision "affected rift growth more substantially" (line 187), and background glaciological stresses are obviously important. So I suggest you change the headline to include reference to at least the iceberg collision event.

Reviewer #3

(Remarks to the Author)

Review of

"Ocean tides trigger ice shelf rift growth and calving"

by Marsh et al., submitted to Nature Communications, July 2024

Overview

Thanks to its hosting of Halley Station, Brunt Ice Shelf is probably one of the best studied in Antarctica. Recent events related to fracture propagation and major iceberg calvings afforded the authors an excellent opportunity to conduct valuable in situ measurements of those changes, augmented with satellite observations, and analyzed with the use of a simplified viscoelastic rheological model. The authors use that analysis to quantify the relative importance of a set of external forcings on rift opening rates. They reach the conclusion that tides played a dominant role in rift evolution from 2019 onwards, culminating in the major calving of 2023, of which the timing was also influenced by the tides.

I believe the in situ observations presented here to be highly valuable in the endeavour to improve understanding of ice-shelf fracture evolution. This value is accentuated by the rarity of such observations. I also think that the authors present a plausible case for the influence of tidal forcing on fracture evolution. I raise questions below about the manner by which this

influence is exerted, and possible implications to the modeling approach. I should note that, while I do make comments on the modeling, I do not have sufficient experience in fracture mechanics to assess the nuances of the assumptions and simplifications that the authors make.

Finally, I found that the manuscript in general is very well written and organized, and the figures are mostly helpful.

Main Issues

Possible effect of MIR: A major dynamic feature of the Brunt Ice shelf is the McDonald Ice Rumples (MIR), of which the location is indicated in Fig. 1c. My understanding is that the contact between the ice shelf and the bathymetric feature at MIR is intermittent (that is why it is not classified as an “ice rise”, where contact would be permanent). The extent of the contact would vary with the vertical movement of the shelf under the influence of tidal forcing. If this understanding is correct, then that raises the following questions with regards to the modeling approach: Are tide-induced stress and glacial stress truly independent, as seems to be the assumption made in the analysis? Relatedly, is it safe to assume glacial stress to be constant?

Tidal effect (L. 244–251): Correlations are mentioned between rift opening rates and tidal amplitudes and their changes. Is it possible to explicitly give the values of those correlations?

Wind effect (e.g., L. 252–254): Is there observational evidence to suggest that wind stress is related to rift opening changes, as in the case of the tidal effect?

Finding C3 and C4: My understanding is that the parameters C3 and C4 are found through the optimization process described under “Model Fitting” (L. 262). In L. 271–274, there seems to be a suggestion that their values can be constrained using observation. Are the values thus derived then used as initial estimates in the least-squares fitting iterations and/or as bounds on the variations of the two parameters during those iterations?

GNSS stations: Figure 1i suggests that GNSS measurements were made near-continuously from 2017 to 2023. Is Figure 1j showing that the locations of the GNSS stations relative to MIR did not change during that period? If yes, how were the locations of the stations maintained fixed as the ice shelf flows? Relatedly, in measuring the GPS rift widths shown in Figure 1i, was the spreading of the ice shelf taken into account?

General: Could an estimate of uncertainty be provided on the reported measurements, such as rift width and its changes at ApRES?

Other Issues, Typos, etc.

L. 30: Plural: “influence[s]”.

L. 41: “experiences semi-diurnal velocity fluctuations of ± 2 metres”; is this supposed to be “vertical” instead of “velocity”?

L. 63–64: “This was followed by a period of stagnation lasting approximately two years (Fig. 1i).” Perhaps it would be worth noting in the text that even during this period, the rift continued to enlarge as measured at the GPS stations closer to the grounding line.

Figure 1: Consider making the colors of the two GNSS measurements more distinct.

Figure Ext. 7: I am not certain what this figure is showing, especially the circular inset.

Version 1:

Reviewer comments:

Reviewer #1

(Remarks to the Author)

I reviewed a previous version of this manuscript and I think the authors have largely addressed my comments.

There are a couple of minor points that I would encourage the authors to consider to make sure the presentation as consistent as possible. I list these below, but they are all relatively minor points related to word choices.

1. Near line 73 (track changes version), the stress is described as a shear stress, but near line 116 the stress is described as tensile. I think the authors mean tensile in this case since near lines 241 the authors refer to Mode I opening stresses.

2. I'm still confused by the Cauchy stress vs depth-averaged deviatoric stress vs depth-averaged resistive stress distinction here. The hydrostatic component of the pressure is unlikely to change over the time scales considered so I'm happy to drop

that and for a one-dimensional model, the resistive stress and deviatoric stress are proportional to within a factor of 2. This makes the comparison with tensile deviatoric stresses (e.g., near line 116 track changes version) confusing. To compare tensile with resistive stress, don't we need to double the tensile stresses (to account for the factor of 2)? Re-reading, I'm starting to think that the authors are actually estimating depth-averaged deviatoric stresses (?) and not resistive stresses, unless the conversion to resistive stress is done with the effective viscosity? Recognizing that the empirical model is probably only accurate to within a factor of 2, it still might be useful to sign post to readers how the conversion between deviatoric stress and resistive stress was done. The glaciological literature is plagued by missing factors of 2 associated with resistive vs deviatoric stresses.

3. I think that the model by Brunt and MacAyeal (2014) may be relevant to the authors discussion as this study discusses the effect of tides on horizontal motion of the Ross Ice Shelf using a linearized approach. Brunt and MacAyeal (2014) use a shallow shelf numerical model, but the linearization approach is similar and I think the conclusions are analogous (although for intact ice!) and might help support the authors results
Brunt KM, MacAyeal DR. Tidal modulation of ice-shelf flow: a viscous model of the Ross Ice Shelf. *Journal of Glaciology*. 2014;60(221):500-508. doi:10.3189/2014JoG13J203

4. More tangential, the recent paper by Rendfrey et al., (2025), uses a statistical approach to infer statistically significant correlation with atmospheric extremes. This is a bigger stretch, but a conclusion from that paper is that temporally sparse data is not able to reliably distinguish between causal correlations and spurious correlations. The fact that higher resolution field results is able to avoid the spurious correlation entanglement points towards how coarse satellite imagery can be successfully combined with field observations to improve predictions.

Rendfrey TS, Bassis J, Pettersen C. Do atmospheric rivers trigger tabular iceberg calving? *Journal of Glaciology*. 2025;71:e8. doi:10.1017/jog.2024.94

5. The final paragraph suggests that iceberg timing can be better predicted from environmental forcing records. However, Brunt is not a typical ice shelf. As a challenge to the authors, would we expect similar results for other ice shelves? Can results from Brunt be generalized with similar empirical coefficients?

Reviewer #3

(Remarks to the Author)

2nd Review of

"Ocean tides trigger ice shelf rift growth and calving"

by Marsh et al., submitted to *Nature Communications*, July 2024

Overview

I thank the authors for their responses to my earlier comments. Below, the authors' responses to my previous comment are in quote marks.

Main Issues

Previous reviewer comment: Possible effect of MIR: A major dynamic feature of the Brunt Ice shelf is the McDonald Ice Rumples (MIR), of which the location is indicated in Fig. 1c. My understanding is that the contact between the ice shelf and the bathymetric feature at MIR is intermittent (that is why it is not classified as an "ice rise", where contact would be permanent). The extent of the contact would vary with the vertical movement of the shelf under the influence of tidal forcing. If this understanding is correct, then that raises the following questions with regards to the modeling approach: Are tide-induced stress and glacial stress truly independent, as seems to be the assumption made in the analysis? Relatedly, is it safe to assume glacial stress to be constant?

"We agree that the terminology around stresses was confusing. All the stresses we are discussing are 'glacial stresses' in the sense that they are acting within the ice, but we have attempted to separate them in order to determine the importance of tides (and wind). These should perhaps be better labelled as a constant background stress (which we have here called glacial), a stress that varies in correlation with the tides (which we have called tidal) and one that varies in correlation with the wind (which we have called wind). Due to the non-linearity of the flow law, oscillating crack behaviour can be related to oscillating driving stresses, but all behaviour is fundamentally controlled by the background average stress (the glacial part). Over long time periods the glacial stress is not constant (for instance it changes between early 2019 and late 2019), but here we focus on short-periods (~ 1 month) where we believe this assumption to be reasonable. We have modified the stress terminology in response to Reviewer 1."

New reviewer comment: Could you please clarify whether stress variability due to different degrees of ice-shelf grounding at MIR, due to tidally-induced vertical motion, is taken into account in the modeling.

Previous reviewer comment: GNSS stations: Figure 1i suggests that GNSS measurements were made near-continuously

from 2017 to 2023. Is Figure 1j showing that the locations of the GNSS stations relative to MIR did not change during that period? If yes, how were the locations of the stations maintained fixed as the ice shelf flows? Relatedly, in measuring the GPS rift widths shown in Figure 1i, was the spreading of the ice shelf taken into account?

"The GNSS stations advect with ice flow. They are generally moving perpendicular to the ice rumpled so their distance does not change substantially. In Figure 1i the spreading of the ice shelf was not taken into account but it is known across other nearby baselines to be around 0.001 /yr. This would account for ~20 m over 6 years at NN00-OO00 and ~60 m over 6 years at TT01-KK00 which is a small percentage of the total signal. We have added some text to explain this."

New reviewer comment: These statements need support. My understanding from observationally-inferred velocity fields is that flow in that part of Brunt Ice Shelf has a strong westerly component, so the stations would not be moving perpendicular to the ice rumpled. Also, flow speeds according to observations are in the high 100s m/yr, so the advection of the stations over the 6-year observation period would be non-negligible (contrary to what Fig. 1j is showing). Please clarify how these factors might have affected estimates of the width of Chasm-1 between 2017 and 2023.

Version 2:

Reviewer comments:

Reviewer #1

(Remarks to the Author)

The authors have addressed all of my comments and I look forward to seeing the published version of this paper.

Reviewer #3

(Remarks to the Author)

Please see attached file.

Response to Reviewers

We thank the reviewers for their detailed and useful comments, in particular Reviewer 1 whose comments have greatly helped us to improve the manuscript by highlighting ways to better present our explanation of stress and improve the terminology. We respond to each reviewer in turn below (reviewer comments in red; response in black):

Reviewer #1 (Remarks to the Author):

This manuscript describes results from a field campaign on the Brunt Ice Shelf showing that rift propagation appears to be, at least partially, triggered by ocean tides and winds. This study falls into a larger cannon of studies that seek short-term environmental triggers of rifting propagation and iceberg calving. This study is the first that I am aware of that demonstrates a link between rift processes and tidal forcing. One of the challenges, however, of this type of study is that it is difficult to generalize: we do not know if the correlation with tides holds more generally for rifts in the Brunt ice shelf, much less other rifts on other ice shelves. This issue is an issue that has been difficult to overcome in many studies, but doesn't detract from the study presented here.

We agree that tides may not be the dominant trigger of calving elsewhere, or across cracks at different orientation to tidal currents (for instance we state "The sensitivity of rifts on other ice shelves to short-term stresses will depend on their specific geometry, including the relative area of unconstrained ice on which tides and wind can act, and the ligament length attaching nascent icebergs to their ice shelves. There is a large variation in tidal range and rate of change of tide across Antarctica, and we would expect large spatial differences in susceptibility to tidally-triggered fracturing.") Nevertheless, here there is an overwhelming tidally synchronous response of the ice shelf, increasing as calving approaches, and it is clear that tidal information may be useful in assessing calving likelihood over the short-term in some situations.

Overall, I think the study is well written, adds to the literature and will be interesting to a wider audience. However, the methodology and assumptions were difficult for me to fully follow: I struggled to understand and convince myself that some of the assumptions described in the methods are appropriate and link the empirical models to physical models. I recognize that there is a balance that needs to be met with not being overly technical so that the manuscript is accessible to a wide audience and adding more specialized jargon. But I think that some more precision in terminology and clarity in the methods might be useful for readers. This is especially true when asking readers to make the link between analogies and derivations which seem based a little bit more on insight and intuition than formal manipulations. I outline some of these issues below, recognizing that it is possible that some of these are just wording issues while others might point to issues with the methodology that need to be addressed.

Thank you. The main aim of this paper is to present these new observational datasets, and put them into the public domain. A complex physical model (3D, viscoelastic, including fracture) is outside of the scope of this paper, and so we have relied on empirical modelling. We do not suggest that our empirical model captures everything, but it acts as a heuristic / qualitative description of what may be occurring. We have tightened up the wording to better explain the technical aspects of the methodology.

1. Cauchy stress, deviatoric stress and all that. To start, the authors use the term "stress" multiple times throughout the study (e.g., line 69, line 74, but also many other places). Stress is, of course, a tensor, but it is treated in the text as if it is a scalar. This is fine so long as the authors are clear that they are referring to a particular component of the stress tensor (or better yet, the traction). I also assume that the "stress" referred to is depth averaged in some sense, but this is also not stated. The more confusing problem is that the authors do not distinguish between Cauchy stress, deviatoric stress and effective stress. In fracture mechanics, the stress driving crack propagation is the Cauchy stress, but flow of ice is controlled by the deviatoric stress. This is an important distinction because the Cauchy stress can be compressive because of the hydrostatic pressure whilst the deviatoric stress remains positive. More specifically, the effective viscosity in the creep flow-law is controlled by the effective deviatoric stress, which is independent of pressure and depends on the sum of the squares of the components of the stress. This is an important distinction because the effective stress cannot be negative and depends on more than a single component of the deviatoric stress tensor. I would typically write the Equation that appears in lines 69 and 74 in terms of the effective stress (which is always positive). All of this is very confusing to

readers, especially since the tidal stress shown in Figure 2 has negative values. One has to be careful (and clear) about absolute values and what is positive and negative! Here the authors would better serve readers by clearly stating what they mean by "stress" in a more precise way (it is a component of the full stress tensor? is it depth-averaged? is it deviatoric stress or a component of the Cauchy stress).

Many thanks. Here we are using 'stress' to mean a scalar depth-averaged longitudinal resistive stress, which is the rift normal stress responsible for ice shelf spreading. We assume ice thickness and density to be spatially uniform and this allows us to consider the ice shelf as a 2D plate without vertical shear. We also only consider one stress direction, the one that is perpendicular to the crack such that the crack is loaded uniaxially, (making an assumption of mode I opening). The resistive stress (R_{xx}) is the Cauchy stress minus the hydrostatic stress (e.g. Van der Veen (1998)). It is unlikely that crack propagation occurs along a perfectly vertical plane as the vertical profile is complicated by temperature and density dependencies plus water pressure, but here we assume uniform resistive stress with depth for simplicity (e.g. Rist et al., 1996). If the other components of the deviatoric stress tensor are small then variation in effective stress is only affected by variation in resistive stress and so it is reasonable to link this stress to the deformation. The 'negative' tidal (resistive) stress then leads to a positive effective stress which is important for the non-linearity in deformation. We do not consider 3D effects or bending, although we admit these may be important, and we assume the ice is constant density, temperature and thickness.

The approach we use here is based on the proposition that when the ice breaks at a particular critical depth, it has broken everywhere and will be visible at the surface. We appreciate that a simple mode 1 crack in one spatial dimension may be an oversimplification, particularly where water pressure acts at depth, but a full 3D simulation of the stress field is outside the scope of this work.

2. Local fracture stress, remote stress and rift opening angle. Putting aside the terminology issues, the Equation in Line 69 is valid for a 1-dimensional visco-elastic material. I don't follow the subsequent assumptions and approximations made to then transform the rheological equation in Line 69 to rift opening angle in Line 78. This seems like an analogy then a derivation, which is fine but should be stated. If we consider uni-axial loading transverse to the rift and ignore all other components of the stress tensor, then it seems like we should be able to write an equation in the form of the Equation on Line 69 with $\epsilon(t)$ denoting the component of strain transverse to the rift. The integral of the strain would be the rift opening/displacement. The rift opening angle used here appears to be defined as the ratio of the width to the length of the rift and this is, at least dimensionally, equivalent to a shear strain. This adds confusion about which component of the stress is considered. It would be helpful to clarify this, but either way, I don't see why the factors of A and E in the Equation in 74 has been dropped. Maybe the ratio is used to scale the rift opening to rift length, but rift width in an elastic material should scale with rift length. Moreover, the rheology is a statement about material properties and has nothing to do with the various stress balances in the far-field or fracture regions. The magnitude of stresses might vary spatially and in the near vs far field, but the rheology is not a statement about the stress balance. Hence, it is not clear to me why we need both equations or why they are written in different forms.

It is correct that we are working in 1 dimension. The reason we use angle instead of strain across the rift is that we wish to compare the different time periods like-for-like, when the crack tip is moving. The crack width is converted to angle to normalise the distance between our instrument (ApRES) and the crack tip. The 'raw' width data in Figure 1 does not do this, but by converting width to opening angle, we adjust for the fact that the measurement point moves further away from the crack tip when propagation occurs. In practice it makes little difference as the angles are so small. Either way, it is not a shear strain but an adjustment to allow a direct comparison between measurements of transverse strain from different time periods with a slightly altered crack geometry.

We have replaced A and E into the second equation for clarity. These had been rolled into the proportionality constants as they are treated as constant over time and space.

3. Fracture mechanics and other stuff. The methodology sketched out in Lines 218-226 argues for a relationship between the local stress near the crack tip and the far-field stress that is based on fracture mechanics. As I understand it, I think the authors are trying to infer the stress near the crack tip based on the far field stress, which is perfectly reasonable for an elastic material. I can follow the scaling presented, but given the relationships in Equations near lines 218-223 and the fact that the rift length is known, why not use the rift length in the analysis? The proportionality is given by a factor that is related to the rift

length and another factor this typically of order unity (often π or some such). At least for an order of magnitude estimate, we could directly estimate the stress associated with different forcing without introducing arbitrary constants. In fact, an alternative interpretation of the analysis is that if we divide both sides of the Equation in line 218 by the square root of the rift half-length, then we find that the remote stress required to drive rift propagation is approximately zero given the critical stress intensity factor from lab measurements. Hence, whenever the Cauchy stress (not deviatoric stress!) is positive, the rift will propagate. This relationship only holds for linear elastic materials, which gets me into my next point.

Here we run in difficulties in trying to accurately model fracture in a non-linear viscoelastic material, which is not fully covered by existing literature. We have not used the rift length as the rift in question is not a simple geometry (and the ice is not a elastic material). The rift also contains melange / sea ice material such that it is difficult to define it's extent, and contrary to our model, the thickness does vary spatially. We have removed some of the text on stress intensities as we are not using LEFM here.

4. Viscoelasticity and Hoff's analogy. As I understand it, the analogy pointed out by the authors holds for linear viscosity and linear elasticity (or power-law elasticity and power-law creep), but not between linear elasticity and power-law creep. In fact the stress singularity associated with power-law creep has been deduced and it depends on the flow-law exponent. (The singularity is exactly the same as for an elastic material if the flow law exponent is unity.). As a consequence, I don't think that the scaling argument that the authors apply follows. Or, at least, more clarification is needed to explain why it works. The way that the authors present the methods make it seem like this is a critical flaw in the methodology or something that needs additional justification. (I apologize if I misunderstood!) However, reading through a few more times I'm not quite sure that the authors analysis really depends on this scaling argument and I wonder if the authors can simply use the rheology presented in lines 69 with some appropriate measure of transverse strain and simplify much of this exposition.

Yes we have decided to remove this as it adds complexity that is better left out in this simple empirical approach.

5. My penultimate concern about the methodology is, perhaps, more of a philosophical issue. The main conclusion is that tides and, to a lesser extent winds, drive rift propagation. The concerns that I raised are all based on the physical model presented, but the authors conclusions are fundamentally based on an empirical correlation with 4 tuned coefficients (5 if the phase of the tide is included) that are fit to the data. However, we can, at the very least, provide ball park estimates of all of the stress terms based on the data collected. The drag associated with tides and winds depends quadratically on the current/wind speed, density of water/air and a drag coefficient. The wind speed is measured. The current speed can be extracted from the same tidal model used to estimate ocean height. The only unknown is the drag coefficient, which can vary over several orders of magnitude. Nonetheless, the authors can ballpark estimate the stress associated with different forcing using a range of published drag coefficients to determine if the relative contributions inferred empirically are broadly consistent with what we expect from a more physical model. If the forcing is broadly consistent, then it is a strong check on the more empirical methodology and might negate my confusion about the methods. Moreover this might even allow us to estimate the role that winds and tides have on other rifts. This would be a significant advance in the field. If the relative magnitudes are not consistent then it implies drag coefficients need to be much larger/smaller than typically inferred or something else is going on that needs additional study. Perhaps the Brunt is unusual or there is missing physics we still need to understand. This is also important to know.

Our empirical measurements are broadly in line with estimates from tidal currents although drag coefficients need to be higher than usually assumed (e.g. a basal drag coefficient of between 0.001 and 0.01 (Gwyther et al., 2015) and current peaking at 0.04 m/s would produce a drag of 0.0008 to 0.008 N/m²).

The reason we have not included these calculations here is due to the additional 2D effects (e.g. moment) which complicate our simple model, and for which we do not have enough data. We don't wish to repeat the work of e.g. Bassis et al., 2008, but instead seek to provide *relative* contributions from different sources. This is possible because of the precision with which we know the crack behaviour over multiple

time scales. We now show correlations with the westerly tidal current from CATS2008 (Padman et al., 2002) in Figure Ext. 6e. The phase agrees well, but the shape of the oscillations still matches more closely with the gradient of the tidal amplitude (Ext. 6d).

6. My final point is that it seems as though the authors are using rift widening as a proxy for rift opening. Rift widening in an elastic material is either due to a change in the stress opening the rift and/or rift propagation. In a viscoelastic material the situation is more complicated because the rift will widen due to ductile flow even with no rift propagation. In an elastic material, rift widening results in stored elastic energy that can be released by rift propagation. In a disco-elastic material, rift widening may dissipate stored elastic energy or respond passively to rift propagation. This confusion may be addressed in the manuscript, but it seems possible that the rapid rift widening stages observed are caused by rift propagation, which stand in contrast with rapid rift widening causing stress accumulation that then results in rift propagation. In other words, the direction of causality isn't clear to me. Can the possibility that rapid rift widening is simply a response to the fact that the rift is or has recently propagated?

No, we are happy that the causality is in this direction. We have 10-minute resolution data and the widening is not as rapid as might be expected from a propagation event. We observe that in the short-term the propagation events have a negligible effect on this opening angle.

Minor comments:

A key finding is that the square of the tidal forcing results in fortnightly forcing. Recent studies, however, have increasingly shown that on ice shelves the flow-law exponent is probably closer to 4. Would this significantly affect the results? Would the fortnightly tidal affect still appear? Does a flow law exponent of 4 help with the tidal phase offset needed in the fit? If the fortnightly forcing is robust, then this seems like a useful prediction of the model based on fundamental principles that is worth further emphasizing.

A flow law exponent of 4 would affect the results in that there would be two tidal terms that introduce a non-linearity ($g^2 * t^2$; and t^4). The fortnightly signal would still exist. It is possible that this dataset could be used to inform the exponent in the flow law, but we are already doing this for A.

Is it possible that the connection with wind speed is related to small movements of the stations of the ice associated with torque from the wind?

No the stations are guyed very tightly and the reflector is a mesh (to avoid catching the wind).

Line 84: This is the results of expanding the cubic, right? Why are the factors of 3 and 6 in the terms dropped? Line 265: This equation includes all of the factors of 3 and 6s that was missing in the main text equation. It might be easier for readers if these equations are presented in a consistent form.

This is just the components of the opening angle (and which stress components they are derived from in the subscript). The factors are not important here as it is not an equation, but a labelling of the different terms.

Line 100: kts—use metric units.

OK.

Why do tides result in a phase shifted stress? This isn't intuitive to me. Is it somehow related to the phase difference between tidal currents that drive propagation and tidal heights?

Yes it is a phase difference between the tide heights and the currents. While there is some uncertainty on the mean current, the tidally induced currents show a direct correlation without a phase shift with the widening rate. We have added a panel to Figure Ext. 6 showing this.

Reviewer #2 (Remarks to the Author):

This is an excellent, interesting and important paper that reports highly detailed observations of rift propagation and iceberg release from Brunt Ice Shelf in Antarctica, and an insightful analysis of the key forces at work. The methods are rigorous and clearly explained and the conclusions justified. Illustrations are excellent.

Thank you.

I see no need for any revision of the text, except for one minor correction. On line 41, there is mention of "velocity fluctuations of plus/minus 2 metres", which should of course be elevation fluctuations, which correspond with velocity fluctuations. It might also be useful to add the units for 'rift opening angle' which I presume are radians (line 72 and Fig 2).

Yes this is a mistake and has been corrected. Angles in radians do not normally require units and I believe this is the case in Nature journals.

The main change that I think might be useful is to the title of the paper. The title only mentions tidal forcing as the trigger for both rift propagation and calving, but it is clear from the text that this is not the whole story. While rising tidal stresses clearly coincided with individual rift propagation events, iceberg collision "affected rift growth more substantially" (line 187), and background glaciological stresses are obviously important. So I suggest you change the headline to include reference to at least the iceberg collision event.

We have chosen not to change the title as while collision is important it did not cause the calving and is a less important finding. There are several other studies showing the importance of iceberg collision (i.e. it is not a new discovery) and here we wish to emphasise the critical importance of ocean tides, which have previously been overlooked.

Reviewer #3 (Remarks to the Author):

Overview

Thanks to its hosting of Halley Station, Brunt Ice Shelf is probably one of the best studied in Antarctica. Recent events related to fracture propagation and major iceberg calvings afforded the authors an excellent opportunity to conduct valuable in situ measurements of those changes, augmented with satellite observations, and analyzed with the use of a simplified viscoelastic rheological model. The authors use that analysis to quantify the relative importance of a set of external forcings on rift opening rates. They reach the conclusion that tides played a dominant role in rift evolution from 2019 onwards, culminating in the major calving of 2023, of which the timing was also influenced by the tides.

I believe the in situ observations presented here to be highly valuable in the endeavour to improve understanding of ice-shelf fracture evolution. This value is accentuated by the rarity of such observations. I also think that the authors present a plausible case for the influence of tidal forcing on fracture evolution. I raise questions below about the manner by which this influence is exerted, and possible implications to the modeling approach. I should note that, while I do make comments on the modeling, I do not have sufficient experience in fracture mechanics to assess the nuances of the assumptions and simplifications that the authors make.

Finally, I found that the manuscript in general is very well written and organized, and the figures are mostly helpful.

Thank you.

Main Issues

Possible effect of MIR: A major dynamic feature of the Brunt Ice shelf is the McDonald Ice Rumples (MIR),

of which the location is indicated in Fig. 1c. My understanding is that the contact between the ice shelf and the bathymetric feature at MIR is intermittent (that is why it is not classified as an “ice rise”, where contact would be permanent). The extent of the contact would vary with the vertical movement of the shelf under the influence of tidal forcing. If this understanding is correct, then that raises the following questions with regards to the modeling approach: Are tide-induced stress and glacial stress truly independent, as seems to be the assumption made in the analysis? Relatedly, is it safe to assume glacial stress to be constant?

We agree that the terminology around stresses was confusing. All the stresses we are discussing are ‘glacial stresses’ in the sense that they are acting within the ice, but we have attempted to separate them in order to determine the importance of tides (and wind). These should perhaps be better labelled as a constant background stress (which we have here called glacial), a stress that varies in correlation with the tides (which we have called tidal) and one that varies in correlation with the wind (which we have called wind). Due to the non-linearity of the flow law, oscillating crack behaviour can be related to oscillating driving stresses, but all behaviour is fundamentally controlled by the background average stress (the glacial part).

Over long time periods the glacial stress is not constant (for instance it changes between early 2019 and late 2019), but here we focus on short-periods (~ 1 month) where we believe this assumption to be reasonable.

We have modified the stress terminology in response to Reviewer 1.

Tidal effect (L. 244–251): Correlations are mentioned between rift opening rates and tidal amplitudes and their changes. Is it possible to explicitly give the values of those correlations?

Correlations are given in Figure Ext. 6.

Wind effect (e.g., L. 252–254): Is there observational evidence to suggest that wind stress is related to rift opening changes, as in the case of the tidal effect?

Yes, the widening rate is increased during strong easterly wind events and appears to correlate with the square of the wind speed. The effect is less than for tides under normal wind speeds seen on the Brunt Ice Shelf.

Finding C3 and C4: My understanding is that the parameters C3 and C4 are found through the optimization process described under “Model Fitting” (L. 262). In L. 271–274, there seems to be a suggestion that their values can be constrained using observation. Are the values thus derived then used as initial estimates in the least-squares fitting iterations and/or as bounds on the variations of the two parameters during those iterations?

The values of all parameters C1-C4 are obtained from the observations via the fitting procedure which is not iterative but provides a minimum least-squares. The ranges to search through are predefined. Because of the non-linearity in ice flow, once C1-C4 are calculated the absolute value for stress can be estimated.

GNSS stations: Figure 1i suggests that GNSS measurements were made near-continuously from 2017 to 2023. Is Figure 1j showing that the locations of the GNSS stations relative to MIR did not change during that period? If yes, how were the locations of the stations maintained fixed as the ice shelf flows? Relatedly, in measuring the GPS rift widths shown in Figure 1i, was the spreading of the ice shelf taken into account?

The GNSS stations advect with ice flow. They are generally moving perpendicular to the ice rumples so their distance does not change substantially. In Figure 1i the spreading of the ice shelf was not taken into account but it is known across other nearby baselines to be around 0.001 /yr. This would account for ~ 20 m over 6 years at NN00-0000 and ~ 60 m over 6 years at TT01-KK00 which is a small percentage of the total signal. We have added some text to explain this.

General: Could an estimate of uncertainty be provided on the reported measurements, such as rift width

and its changes at ApRES?

The uncertainty on the ApRES data ($\sim \pm 2$ mm) is substantially lower than the daily tidal changes (± 50 - 100 mm) within a day. The GPS data has a higher uncertainty but here is only used for context to the longer-term behaviour. We have added this in the Appendix.

Other Issues, Typos, etc.

L. 30: Plural: "influence[s]".

Change.

L. 41: "experiences semi-diurnal velocity fluctuations of ± 2 metres"; is this supposed to be "vertical" instead of "velocity"?

Yes.

L. 63-64: "This was followed by a period of stagnation lasting approximately two years (Fig. 1i)." Perhaps it would be worth noting in the text that even during this period, the rift continued to enlarge as measured at the GPS stations closer to the grounding line.

OK.

Figure 1: Consider making the colors of the two GNSS measurements more distinct.

We have kept the colour scheme the same because the right-hand y-axis in panel (i) needs to match both colours (so they need to be similar). They are hopefully distinct enough that it is clear which is which.

Figure Ext. 7: I am not certain what this figure is showing, especially the circular inset.

We have removed this figure.

Reviewer #1 (Remarks to the Author):

I reviewed a previous version of this manuscript and I think the authors have largely addressed my comments.

There are a couple of minor points that I would encourage the authors to consider to make sure the presentation as consistent as possible. I list these below, but they are all relatively minor points related to word choices.

1. Near line 73 (track changes version), the stress is described as a shear stress, but near line 116 the stress is described as tensile. I think the authors mean tensile in this case since near lines 241 the authors refer to Mode I opening stresses.

We have made the discussion of stress in the appendix substantially clearer. We hope it is now clear how we've reached a model where the elastic and viscous behaviours are functions of the same stress.

2. I'm still confused by the Cauchy stress vs depth-averaged deviatoric stress vs depth-averaged resistive stress distinction here. The hydrostatic component of the pressure is unlikely to change over the time scales considered so I'm happy to drop that and for a one-dimensional model, the resistive stress and deviatoric stress are proportional to within a factor of 2. This makes the comparison with tensile deviatoric stresses (e.g., near line 116 track changes version) confusing. To compare tensile with resistive stress, don't we need to double the tensile stresses (to account for the factor of 2)? Re-reading, I'm starting to think that the authors are actually estimating depth-averaged deviatoric stresses (?) and not resistive stresses, unless the conversion to resistive stress is done with the effective viscosity? Recognizing that the empirical model is probably only accurate to within a factor of 2, it still might be useful to sign post to readers how the conversion between deviatoric stress and resistive stress was done. The glaciological literature is plagued by missing factors of 2 associated with resistive vs deviatoric stresses.

Yes we do think it is now clearer to refer to deviatoric stresses throughout (although we think the use of resistive stress is useful for understanding what we are doing, so we have left it in the appendix derivation). We have further elaborated on our definitions, derivations and assumptions in the appendix. We also now use E' in the main paper, which differs from E by a factor of between 1.5 and 2 (depending on Poisson's ratio) and which slightly modifies our overall calculated stress threshold. This means we have replotted Figures 2 and 3 (and the extended figures 2 and 7).

3. I think that the model by Brunt and MacAyeal (2014) may be relevant to the authors discussion as this study discusses the effect of tides on horizontal motion of the Ross Ice Shelf using a linearized approach. Brunt and MacAyeal (2014) use a shallow shelf numerical model, but the linearization approach is similar and I think the conclusions are analogous (although for intact ice!) and might help support the authors results

Brunt KM, MacAyeal DR. Tidal modulation of ice-shelf flow: a viscous model of the Ross Ice Shelf. *Journal of Glaciology*. 2014;60(221):500-508. doi:10.3189/2014JoG13J203

Thank you, we have added some additional references here, including this one.

4. More tangential, the recent paper by Rendfrey et al., (2025), uses a statistical approach to infer statistically significant correlation with atmospheric extremes. This is a bigger stretch, but a conclusion from that paper is that temporally sparse data is not able to reliably distinguish between causal correlations and spurious correlations. The fact that higher resolution field results is able to avoid the spurious correlation entanglement points towards how coarse satellite imagery can be successfully combined with field observations to improve predictions.

Rendfrey TS, Bassis J, Pettersen C. Do atmospheric rivers trigger tabular iceberg calving? *Journal of Glaciology*. 2025;71:e8. doi:10.1017/jog.2024.94

This is also a valuable reference that I was previously unaware of. In the dataset we present here atmospheric events did contribute to crack growth on several occasions (in particular when they aligned with spring tides), but tides alone (plus the evolving glaciological situation) appear to have been responsible for the final calving of A-81, with no atmospheric contribution.

5. The final paragraph suggests that iceberg timing can be better predicted from environmental forcing records. However, Brunt is not a typical ice shelf. As a challenge to the authors, would we expect similar results for other ice shelves? Can results from Brunt be generalized with similar empirical coefficients?

A similarly high-resolution field dataset should be acquired on a more 'standard' ice shelf if possible, but I think it is feasible to generalise if crack geometry is well known.

Reviewer #3 (Remarks to the Author):

2nd Review of

“Ocean tides trigger ice shelf rift growth and calving“

by Marsh et al., submitted to *Nature Communications*, July 2024

Overview

I thank the authors for their responses to my earlier comments. Below, the authors' responses to my previous comment are in quote marks.

Main Issues

Previous reviewer comment: Possible effect of MIR: A major dynamic feature of the Brunt Ice shelf is the McDonald Ice Rumples (MIR), of which the location is indicated in Fig. 1c. My understanding is that the contact between the ice shelf and the bathymetric feature at MIR is intermittent (that is why it is not classified as an “ice rise”, where contact would be permanent). The extent of the contact would vary with the vertical movement of the shelf under the influence of tidal forcing. If this understanding is correct, then that raises the following questions with regards to the modeling approach: Are tide-induced stress and glacial stress truly independent, as seems to be the assumption made in the analysis? Relatedly, is it safe to assume glacial stress to be constant?

"We agree that the terminology around stresses was confusing. All the stresses we are discussing are ‘glacial stresses’ in the sense that they are acting within the ice, but we have attempted to separate them in order to determine the importance of tides (and wind). These should perhaps be better labelled as a constant background stress (which we have here called glacial), a stress that varies in correlation with the tides (which we have called tidal) and one that varies in correlation with the wind (which we have called wind). Due to the non-linearity of the flow law, oscillating crack behaviour can be related to oscillating driving stresses, but all behaviour is fundamentally controlled by the background average stress (the glacial part). Over long time periods the glacial stress is not constant (for instance it changes between early 2019 and late 2019), but here we focus on short-periods (~ 1 month) where we believe this assumption to be reasonable. We have modified the stress terminology in response to Reviewer 1."

New reviewer comment: Could you please clarify whether stress variability due to different degrees of ice-shelf grounding at MIR, due to tidally-induced vertical motion, is taken into account in the modeling.

The terminology ‘ice rumples’ is incorrect here. The McDonald Ice Rumples is an ice rise (see for example Humbert et al., 2009). We are not able to assign the cause of the tidally-induced variability using this simple 1d model. It is possible though that it is due to variability in grounding at the MIR. However, the reason we think this is less likely is due to the phase of tide height relative to the phase of the crack widening. Under a grounding / ungrounding mechanism it would seem sensible that the greatest width would be expected to correlate with highest tide (i.e. when grounding is at a minimum). Here we see that width correlates more strongly with easterly ocean current amplitude which is around 6 hours out of phase with the tide height. We are not saying this is definitive, and certainly a more comprehensive model would provide better insight here.

We do show that the ‘tide-induced’ stresses and ‘glacial’ stresses are NOT independent though and that the tides enhance the overall stress in a non-linear way. We feel this is a terminology issue that we tried to rectify in the previous review but have tried again to make this clearer.

Previous reviewer comment: GNSS stations: Figure 1i suggests that GNSS measurements were made near-continuously from 2017 to 2023. Is Figure 1j showing that the locations of the GNSS stations relative to MIR did not change during that period? If yes, how were the locations of the stations

maintained fixed as the ice shelf flows? Relatedly, in measuring the GPS rift widths shown in Figure 1i, was the spreading of the ice shelf taken into account?

"The GNSS stations advect with ice flow. They are generally moving perpendicular to the ice rumpled so their distance does not change substantially. In Figure 1i the spreading of the ice shelf was not taken into account but it is known across other nearby baselines to be around 0.001 /yr. This would account for ~20 m over 6 years at NN00-OO00 and ~60 m over 6 years at TT01-KK00 which is a small percentage of the total signal. We have added some text to explain this."

New reviewer comment: These statements need support. My understanding from observationally-inferred velocity fields is that flow in that part of Brunt Ice Shelf has a strong westerly component, so the stations would not be moving perpendicular to the ice rumpled. Also, flow speeds according to observations are in the high 100s m/yr, so the advection of the stations over the 6-year observation period would be non-negligible (contrary to what Fig. 1j is showing). Please clarify how these factors might have affected estimates of the width of Chasm-1 between 2017 and 2023.

The ice flow is almost exactly in a westerly direction $\sim 265^\circ$ (and is hundreds of metres per year). However, the crack is in a north-south direction and the MIR is almost exactly north of our instruments (TT01 and NN00 on the eastern side of the crack are slightly south-east, KK00 and OO00 on the western side are slightly south-west) – there is a north arrow in panel 1b. This geometry means that ice advection moves our instruments (and Chasm-1) orthogonally with respect to the MIR.

All the instruments advect by ~ 4 -5 km over this period in terms of absolute position, but their distance to the MIR changes with opposite sign (TT01 and NN00 get closer while KK00 and OO00 get further away) such that the centre of the baselines (which is what is represented here) do not move substantially. The centre of the TT01-KK00 baseline is 5.2 km from the MIR in 2017 and 5.6 km from the MIR in 2022 (+400m). The centre of the NN00-OO00 baseline is 21.2 km from the MIR in 2017 and 20.5 km from the MIR in 2022 (-700m). These changes ARE negligible relative to the crack growth of 20+ km and in no way do they influence the calculated widths which are produced from differencing the positions of the GNSS sites.

The primary aim of panel j is to show where the GPS baselines cross the crack relative to the advancing tip position. We have now modified this panel to account for these small changes in distance caused by the advection (although it is barely noticeable). The full GNSS data is available as a DOI with the paper if it is of interest to readers.

3rd Review of

"Ocean tides trigger ice shelf rift growth and calving"

by Marsh et al., submitted to Nature Communications, July 2024

Previous reviewer comment: Possible effect of MIR: A major dynamic feature of the Brunt Ice shelf is the McDonald Ice Rumples (MIR), of which the location is indicated in Fig. 1c. My understanding is that the contact between the ice shelf and the bathymetric feature at MIR is intermittent (that is why it is not classified as an "ice rise", where contact would be permanent). The extent of the contact would vary with the vertical movement of the shelf under the influence of tidal forcing. If this understanding is correct, then that raises the following questions with regards to the modeling approach: Are tide-induced stress and glacial stress truly independent, as seems to be the assumption made in the analysis? Relatedly, is it safe to assume glacial stress to be constant? "We agree that the terminology around stresses was confusing. All the stresses we are discussing are 'glacial stresses' in the sense that they are acting within the ice, but we have attempted to separate them in order to determine the importance of tides (and wind). These should perhaps be better labelled as a constant background stress (which we have here called glacial), a stress that varies in correlation with the tides (which we have called tidal) and one that varies in correlation with the wind (which we have called wind). Due to the non-linearity of the flow law, oscillating crack behaviour can be related to oscillating driving stresses, but all behaviour is fundamentally controlled by the background average stress (the glacial part). Over long time periods the glacial stress is not constant (for instance it changes between early 2019 and late 2019), but here we focus on short-periods (~ 1 month) where we believe this assumption to be reasonable. We have modified the stress terminology in response to Reviewer 1."

New reviewer comment: Could you please clarify whether stress variability due to different degrees of ice-shelf grounding at MIR, due to tidally-induced vertical motion, is taken into account in the modeling.

The terminology 'ice rumples' is incorrect here. The McDonald Ice Rumples is an ice rise (see for example Humbert et al., 2009). We are not able to assign the cause of the tidally-induced variability using this simple 1d model. It is possible though that it is due to variability in grounding at the MIR. However, the reason we think this is less likely is due to the phase of tide height relative to the phase of the crack widening. Under a grounding / ungrounding mechanism it would seem sensible that the greatest width would be expected to correlate with highest tide (i.e. when grounding is at a minimum). Here we see that width correlates more strongly with easterly ocean current amplitude which is around 6 hours out of phase with the tide height. We are not saying this is definitive, and certainly a more comprehensive model would provide better insight here.

We do show that the 'tide-induced' stresses and 'glacial' stresses are NOT independent though and that the tides enhance the overall stress in a non-linear way. We feel this is a terminology issue that we tried to rectify in the previous review but have tried again to make this clearer.

Final reviewer comment: That MIR is an ice rise, rather than rumples, is new information to me. It seems to be new information to the authors as well, as in my initial comments I specifically mentioned that, to my knowledge, MIR is not an ice rise (see above), and the authors at the time did not contradict that statement. I have no further comment.

Previous reviewer comment: GNSS stations: Figure 1i suggests that GNSS measurements were made near-continuously from 2017 to 2023. Is Figure 1j showing that the locations of the GNSS stations relative to MIR did not change during that period? If yes, how were the locations of the stations maintained fixed as the ice shelf flows? Relatedly, in measuring the GPS rift widths shown in Figure 1i, was the spreading of the ice shelf taken into account?

"The GNSS stations advect with ice flow. They are generally moving perpendicular to the ice rumples so their distance does not change substantially. In Figure 1i the spreading of the ice shelf was not taken into account but it is known across other nearby baselines to be around 0.001 /yr. This would account for ~20 m over 6 years at NN00-0000 and ~60 m over 6 years at TT01-KK00 which is a small percentage of the total signal. We have added some text to explain this."

New reviewer comment: These statements need support. My understanding from observationally-inferred velocity fields is that flow in that part of Brunt Ice Shelf has a strong westerly component, so the stations would not be moving perpendicular to the ice rumples. Also, flow speeds according to observations are in the high 100s m/yr, so the advection of the stations over the 6-year observation period would be non-negligible (contrary to what Fig. 1j is showing). Please clarify how these factors might have affected estimates of the width of Chasm-1 between 2017 and 2023.

The ice flow is almost exactly in a westerly direction $\sim 265^\circ$ (and is hundreds of metres per year). However, the crack is in a north-south direction and the MIR is almost exactly north of our instruments (TT01 and NN00 on the eastern side of the crack are slightly south-east, KK00 and O000 on the western side are slightly south-west) – there is a north arrow in panel 1b. This geometry means that ice advection moves our instruments (and Chasm-1) orthogonally with respect to the MIR.

All the instruments advect by $\sim 4\text{-}5$ km over this period in terms of absolute position, but their distance to the MIR changes with opposite sign (TT01 and NN00 get closer while KK00 and O000 get further away) such that the centre of the baselines (which is what is represented here) do not move substantially. The centre of the TT01-KK00 baseline is 5.2 km from the MIR in 2017 and 5.6 km from the MIR in 2022 (+400m). The centre of the NN00-O000 baseline is 21.2 km from the MIR in 2017 and 20.5 km from the MIR in 2022 (-700m). These changes ARE negligible relative to the crack growth of 20+ km and in no way do they influence the calculated widths which are produced from differencing the positions of the GNSS sites.

The primary aim of panel j is to show where the GPS baselines cross the crack relative to the advancing tip position. We have now modified this panel to account for these small changes in distance caused by the advection (although it is barely noticeable). The full GNSS data is available as a DOI with the paper if it is of interest to readers.

Final reviewer comment: Thank you for clarifying what is being shown in panel j. I have no further comment.